# Predictors of Health-Workforce Job Satisfaction in Primary Care Settings: Insights from a Cross-Sectional Multi-Country Study in Eight African Countries

**DOI:** 10.3390/ijerph22071108

**Published:** 2025-07-15

**Authors:** Samuel Muhula, Yvonne Opanga, Saida Kassim, Lazarus Odeny, Richard Zule Mbewe, Beverlyne Akoth, Mable Jerop, Lizah Nyawira, Ibrahima Gueye, Richard Kiplimo, Thom Salamba, Jackline Kiarie, George Kimathi

**Affiliations:** 1Amref Health Africa, Headquarters, Nairobi P.O. Box 27691-00506, Kenya; samuel.muhula@amref.org (S.M.); lazarus.odeny@amref.org (L.O.); richard.mbewe@amref.org (R.Z.M.); beverlyne.akoth@amref.org (B.A.); mable.jerop@amref.org (M.J.); lizahnm@gmail.com (L.N.); ibrahima.gueye@amref.org (I.G.); richard.kiplimo@amref.org (R.K.); jackline.kiarie@amref.org (J.K.); george.kimathi@amref.org (G.K.); 2Amref Health Africa, Kenya Country Office, Nairobi P.O. Box 30125-00100, Kenya; yvonne.opanga@amref.org; 3Amref Health Africa, Malawi Country Office, Lilongwe P.O. Box 30768, Malawi; salamba.salamba@amref.org

**Keywords:** job satisfaction, health workforce, primary healthcare, sub-Saharan Africa

## Abstract

Job satisfaction in sub-Saharan Africa is crucial as it directly impacts employee productivity, retention, and overall economic growth, fostering a motivated workforce that drives regional development. In sub–Saharan Africa, poor remuneration, limited professional development opportunities, and inadequate working conditions impact satisfaction. This study examined job-satisfaction predictors among health workers in primary healthcare settings across eight countries: Ethiopia, Kenya, Malawi, Senegal, South Sudan, Tanzania, Uganda, and Zambia. A cross-sectional study surveyed 1711 health workers, assessing five dimensions: employer–2employee relationships, remuneration and recognition, professional development, physical work environment, and supportive supervision. The study was conducted from October 2023 to March 2024. The job-satisfaction assessment tool was adopted from a validated tool originally developed for use in low-income healthcare settings. The tool was reviewed by staff from all the country offices to ensure contextual relevance and organization alignment. The responses were measured on a five-point Likert scale: 0: Not applicable, 1: Very dissatisfied, 2: Dissatisfied, 3: Neutral, 4: Satisfied, and 5: Very satisfied. The analysis employed descriptive and multivariable regression methods. Job satisfaction varied significantly by country. Satisfaction with the employer–employee relationship was highest in Zambia (80%) and lowest in Tanzania (16%). Remuneration satisfaction was highest in Senegal (63%) and Zambia (49%), while it was very low in Malawi (9.8%) and Ethiopia (2.3%). Overall, 44% of respondents were satisfied with their professional development, with Uganda leading (62%) and Ethiopia having the lowest satisfaction level (29%). Satisfaction with the physical environment was at 27%, with Uganda at 40% and Kenya at 12%. Satisfaction with supervisory support stood at 62%, with Zambia at 73% and Ethiopia at 30%. Key predictors of job satisfaction included a strong employer–employee relationships (OR = 2.20, *p* < 0.001), fair remuneration (OR = 1.59, *p* = 0.002), conducive work environments (OR = 1.71, *p* < 0.001), and supervisory support (OR = 3.58, *p* < 0.001. Improving the job satisfaction, retention, and performance of health workers in sub-Saharan Africa requires targeted interventions in employer–employee relationships, fair compensation, supportive supervision, and working conditions. Strategies must be tailored to each country’s unique challenges, as one-size-fits-all solutions may not be effective. Policymakers should prioritize these factors to build a motivated, resilient workforce, with ongoing research and monitoring essential to ensure sustained progress and improved healthcare delivery.

## 1. Introduction

Job satisfaction reflects an employee’s overall attitude towards their job, including their feelings about various aspects of the job or an emotional response defining the degree to which people like their jobs, which may range from extreme satisfaction to extreme dissatisfaction [1]. It significantly influences job performance, commitment, absenteeism, retention, and turnover rates, with dissatisfied health workers often seeking employment out of their speciality [2,3,4].

Health workers are the cornerstone of effective health systems, playing a pivotal role in achieving Universal Health Coverage (UHC) and delivering quality primary healthcare (PHC) [5]. In low- and middle-income countries (LMICs), job satisfaction among health workers has been identified as a key determinant of their performance, retention, and overall productivity [6]. This is because a satisfied workforce is more likely to provide better patient care, foster trust and ensure the continuity of care within communities. The significance of job satisfaction has garnered increasing attention, particularly in sub-Saharan Africa, where healthcare systems face numerous challenges, including workforce shortages and high turnover rates. Improving health-workforce job satisfaction in resource-limited settings is essential for addressing systemic health challenges and ensuring the sustainability of health services.

Globally, job satisfaction among healthcare workers is influenced by a complex interplay of factors such as working conditions, the organizational environment, job stress, role conflict and ambiguity, role perception and content, and organizational and professional commitment [7]. Studies have consistently shown that factors such as fair remuneration, supportive supervision, adequate resources, and opportunities for career advancement are strongly associated with higher levels of job satisfaction [8]. Recent Gallup statistics on job satisfaction indicated that a substantial proportion of the world’s 1 billion full-time workers are disengaged and experiencing declining overall well-being. Specifically, 41% of employees are stressed, with one in five experiencing loneliness, half watching for or actively seeking a new job, and one in four experiencing burnout either “very often” or “always” [9]. Europeans are unhappier with their workplaces than workers in any other region, with only 14% of European employees engaged at work, a figure that is seven percentage points lower than the global average (21%) and nineteen (19) points lower than the U.S. and Canada (33%) [9].

The African context presents unique challenges to health-worker job satisfaction. Frequent infectious-disease outbreaks place significant strain on already overburdened healthcare systems, impacting workers’ well-being [10,11]. Less equipped facilities, limited access to essential medicines and supplies, and inadequate infrastructure such as electricity and water, further hinder job satisfaction [12]. In addition, many African countries face severe workforce shortages, leading to heavy workloads and burnout. A study by Muthuri and colleagues conducted in the East Africa Community (EAC) indicates that there are individual, organizational/structural, and societal determinants of healthcare workers’ motivation. The systematic review highlighted barriers reported by the health workforce which included a lack of or inadequate monetary support, favoritism, a critical shortage of skilled healthcare professionals leading to a heavy workload, and unrealistic expectations from management and government. Another study reported that 82.3% of respondents in Tanzania were satisfied with their jobs, compared to 71.0% in Malawi, and 52.1% in South Africa. In all three countries, health workers were most satisfied with their job’s variety and the opportunity to fully utilize their abilities [13].

While low salaries, a lack of benefits, lack of recognition, and unsafe working environments drive high attrition rates [14,15], these challenges present various opportunities that can be harnessed. There is notable willingness among health workers to further develop skills and knowledge and a proactive search for solutions to enhance stock-outs of drugs and other medical devices. There are also motivational factors to improve the quality of care [14]. Non-financial incentives and human-resources management tools play an important role in motivating health professionals. Acknowledging and addressing professional goals, such as career development, recognition and improving their skillset, can uphold and strengthen the professional ethos of health workers [16]. However, little comparative evidence across sub-Saharan Africa countries that simultaneously examines multiple satisfaction domains exists—a gap that this study addresses.

Between October 2023 and March 2024, a multi-country cross-sectional study was conducted across eight African countries including Ethiopia, Kenya, Malawi, Senegal, South Sudan, Tanzania, Uganda, and Zambia. The eight countries are a representation of the different African regions within sub-Saharan Africa including west, east, and southern Africa. These countries were selected based on their diversity in contexts including sociopolitical and economic. Despite these variations, the eight countries share common challenges among the health workforce including resource constraints, shortages in the health workforce, and inequitable access to healthcare services, driven largely by the differences in the urban and rural populations [17,18].

Ethiopia, categorized as a low-income country, is characterized by a predominantly rural population that relies heavily on agriculture. While the country has demonstrated improvement in key health indicators, literacy levels remain low mostly among women [19]. The national health system is organized around a three-tier structure, with the Health Extension Program driving primary-care service delivery through over 40,000 health extension workers deployed in communities [18,20].

Kenya is categorized as a lower-middle-income country characterized by a rapidly urbanizing population and a relatively diversified economy. There are regional and socioeconomic disparities in relation to healthcare and education access as well as income levels [17]. The health system is devolved across 47 county governments where service delivery responsibilities are delegated. The healthcare network includes community health promoters at the community level, dispensaries, health centers, and referral hospitals both at national and subnational levels. The current healthcare reforms include the establishment of primary-care networks to drive primary healthcare and address chronic challenges faced by the system including periodic health-worker strikes and resource constraints [18,21].

Malawi ranks among the poorest countries globally, with more than 70% of its rural population living below the poverty line [19]. The rural population experiences high levels of undernutrition and food insecurity. The health system is structured around district-level service delivery which heavily relies on donor funding. While community health workers drive primary-care services which are widely utilized, the system struggles with health-workforce shortages, drug stockouts, and inadequate infrastructure [18,22].

Senegal is a lower-middle-income country with a growing economy anchored in agriculture, fisheries, and mining. Despite improvements in health and education, urban rural divides in healthcare access persist [17]. The health system follows a pyramidal structure, comprising health posts, centers, and hospitals. Senegal has implemented an advanced universal health-coverage scheme (Couverture Maladie Universelle) and community health strategies to enhance service delivery [18,23].

South Sudan is recognized as the world’s youngest nation. Its health system has been weakened by chronic conflict, political instability, and severe humanitarian challenges. Conflicts have driven the majority of the population to extreme poverty with minimal access to basic services [24]. The health system is underdeveloped, heavily reliant on international aid, and faces chronic shortages of trained personnel and functional health facilities. However, health-services delivery is predominantly provided by non-governmental and faith-based organizations [18,25].

Tanzania has sustained moderate economic growth in recent years but continues to grapple with widespread poverty and a high dependency ratio [17]. The health system is decentralized and structured to provide primary healthcare services through dispensaries and health centers. Community health workers play a pivotal role in delivering preventive and promotive services. Persistent challenges include health-workforce shortages, underfunding, and weak referral systems [18,26].

Uganda has a young and rapidly growing population, with over 75% under the age of 30. While economic performance has improved, poverty and inequality remain significant, particularly in rural regions [19]. The health system is organized into national, regional, district, and community levels. Village Health Teams (VHTs) serve as the first point of contact for health services in many communities. Despite gains in immunization and maternal health, disparities in service quality and access remain [18,27].

Zambia is a lower-middle-income country with substantial mineral wealth, though poverty levels remain high, especially in rural areas [17]. The health system is structured into primary, secondary, and tertiary levels, with an expanding cadre of community health assistants who support service delivery at the grassroots level. While the country has made strides in strengthening health infrastructure and services, it continues to face funding gaps, workforce shortages, and inequities in rural health access [18,28].

The study aimed to inform key strategic directions, including strengthening a fit-for-purpose health workforce for improved skills and productivity. The findings of the study served as a baseline assessment to inform targeted health-workforce interventions, recognizing that each country in sub-Saharan Africa faces unique challenges that require tailored solutions. This study contributes to a growing body of the literature evaluating health-workforce motivation and satisfaction in low- and middle-income countries (LMICs), particularly in sub-Saharan Africa, where human resources for health (HRH) shortages remain a major bottleneck to achieving universal health coverage (UHC) and Sustainable Development Goal 3 [29]. Previous publications in this domain have largely focused on small-scale or country-specific studies, often using qualitative designs or limited sample sizes [30]. Our multi-country, quantitative approach provides a broader comparative perspective, enabling policy-relevant benchmarking across eight countries with differing health-system capacities and reform trajectories.

Governments of the study countries, including Ethiopia, Kenya, Malawi, Senegal, South Sudan, Tanzania, Uganda, and Zambia have recognized health-worker motivation and retention as strategic pillars within their national health-sector strategic plans and HRH policies. For example, Kenya’s Human Resources for Health Strategic Plan [31] emphasizes improving working conditions and incentives for primary healthcare workers; similarly, Uganda’s Health Workforce Strategy (2020–2030) includes components for enhancing supportive supervision and career progression [32]. These strategies align with the WHO’s Global Strategy on Human Resources for Health: Workforce 2030, which underscores job satisfaction as a critical determinant of workforce sustainability and equitable health-service delivery [33].

The relevance of the study is further reinforced by the crucial need to strengthen primary healthcare (PHC) systems in sub-Saharan Africa. As countries shift toward UHC- and PHC-oriented models, frontline health workers must be adequately supported to ensure the continuity of care, improved health outcomes, and community trust [34]. By identifying clear predictors of satisfaction—such as fair remuneration, strong supervisory relationships, and a supportive work environment—this research provides actionable evidence for health-system leaders and donors. These findings can inform targeted investments, particularly through bilateral and multilateral support frameworks like the Global Financing Facility (GFF), PEPFAR, and the Global Fund, all of which increasingly emphasize health-system strengthening.

In addition, by contrasting results with patterns observed in high-income countries, where satisfaction tends to be driven by professional autonomy, career flexibility, and workload balance [35], this study highlights contextual distinctions that are vital for policy design. A one-size-fits-all approach is unlikely to work; therefore, country-level interventions should be tailored to existing health-system bottlenecks, budgetary constraints, and labor-market dynamics. The study’s potential impact lies in its capacity to guide reforms that not only reduce health-worker attrition but which also enhance productivity, quality of care, and, ultimately, health equity across the region.

By identifying key factors such as employer–employee relationships, fair remuneration, supportive supervision, and conducive working environments, the study provides evidence-based insights that can guide policy and decision-making. These findings are crucial for policymakers to prioritize specific areas of improvement to foster a motivated, resilient health workforce, which is essential for achieving sustained progress in healthcare delivery. The study also lays the groundwork for future interventions aimed at addressing workforce challenges and ensuring that healthcare systems in the region are equipped to meet the growing demands of the population. Amref Health Africa assessed the levels and predictors of job satisfaction among health workers in primary healthcare settings across these eight countries.

## 2. Materials and Methods

### 2.1. Study Design

The baseline evaluation employed a cross-sectional study design that surveyed 1711 healthcare workers from 8 countries where Amref implements health-workforce interventions. The countries were chosen to represent a diverse cross-section of sub-Saharan Africa, encompassing eastern, western, and southern African contexts, rather than being selected purely for convenience based on project coverage. By including countries from different regions, the study aims to capture a broader range of health-workforce challenges and job-satisfaction indicators, offering a more comprehensive understanding of the varying dynamics across the region. This approach allows for the identification of region-specific factors while also enabling cross-country comparisons to inform tailored interventions that can be adapted to each country’s unique context. Thus, the inclusion of these countries was a strategic decision to ensure that the findings are relevant and actionable across diverse sub-Saharan African settings. The quantitative methods to assess job satisfaction across five dimensions that were utilized are as follows: employee–employer relationships, remuneration and recognition, professional development, the physical environment and facilities, and supportive supervision.

### 2.2. Study Sites

This study presents the findings from eight African countries—Ethiopia, Kenya, Malawi, Senegal, South Sudan, Tanzania, Uganda, and Zambia. The countries are in the west, east and south of Africa; Figure 1 shows the geographical locations of these countries.

In Ethiopia, data was collected from four regional states—Oromia, Amhara, Afar, and the South Ethiopia regional states and Addis Ababa city administration. The country’s health workforce heavily relies on Health Extension Workers (HEWs), who form the backbone of its primary healthcare system by delivering essential services at the community level, particularly in preventive and promotive care. This program has significantly enhanced access to basic healthcare in rural areas. Ethiopia prioritizes primary healthcare with a strong emphasis on disease prevention and community-based interventions [36,37]. However, challenges include a shortage of trained healthcare professionals, the uneven distribution of workers between urban and rural areas, limited professional-development opportunities, and a high population-to-health-worker ratio, all of which strain the system and impact job satisfaction and service delivery [38].

In Kenya, the study was conducted in eight counties—Nairobi, Narok, Vihiga, Siaya, Nyeri, West Pokot, Samburu, and Tharaka Nithi. These counties were purposively selected to represent diverse geographical settings, including urban, rural, arid, and semi-arid regions, and to address unique Primary Health Care (PHC) challenges and the presence of Amref Health Africa programs. Kenya operates a decentralized healthcare system, with county governments playing a central role in service delivery [39]. The country has a diverse health workforce, including doctors, nurses, clinical officers, and community health volunteers (CHVs), but faces inequitable distribution, particularly in rural and marginalized areas. A robust private-health sector creates competitive opportunities, often contributing to attrition in the public sector. Despite government investments in human resources for health, such as expanding training institutions and increasing healthcare worker training in rural areas, challenges remain in absorbing trained professionals into the workforce, leading to unemployment among qualified health workers [40].

In Uganda, data was collected from the five districts of Iganga, Mayuge, Bugiri, Namayinga, and Pader. Like Kenya, Uganda operates a decentralized health system with a strong focus on community-based healthcare delivery [39]. The country has made progress in addressing workforce shortages by scaling up training programs, particularly for frontline health workers, to improve the detection of, reporting of, and response to disease epidemics [41]. However, significant challenges persist, including widespread workforce shortages, high absenteeism rates, and limited access to essential medical supplies, all of which negatively impact job satisfaction [42]. To address these issues, Uganda has adopted task-shifting strategies, enabling community health workers (CHWs) to take on expanded roles in healthcare delivery [43,44].

Tanzania’s health workforce emphasizes task sharing and integrating CHWs to address acute shortages of skilled professionals [43]. The government has prioritized workforce development in its national strategies, but challenges persist, particularly in rural areas. Issues such as low salaries, limited opportunities for promotion and career development, and inadequate training programs negatively affect workforce morale and retention [45]. While the decentralization of health services has improved community access, it has also increased administrative burdens on health workers. Data was collected from the nine regions of Morogoro, Tanga, Mara, Songwe, Lindi, Tabora, Singida, Kaskazini Unguja, and Kaskazini Pemba.

In Malawi, data was collected from six districts—Mchinji, Chitipa, Karonga, Salima, Mangochi, and Chikwawa. The country has one of the world’s lowest health-worker densities, with critical shortages of skilled professionals such as doctors and nurses [46]. To address this, the government introduced initiatives like the Emergency Human Resources Program to increase workforce numbers and improve distribution. However, low salaries, heavy workloads, and limited access to essential equipment and supplies continue to undermine job satisfaction. International aid and partnerships, including support from the UK Department for International Development (DfID) and the Norwegian Agency for Development Cooperation (NORAD), play a vital role in training and retaining health workers [47].

In Zambia, data was collected from four provinces and districts—Central (Kabwe), Copperbelt (Kitwe), Luapula (Mwense), and Eastern (Sinda). The country’s health workforce has grown due to increased investments in training institutions and partnerships with global-health initiatives. However, high attrition rates remain a challenge, particularly in rural and remote areas. Contributing factors include inadequate remuneration, limited career-advancement opportunities, and insufficient infrastructure. The introduction of community health assistants has been a significant innovation, extending care to underserved areas and alleviating the workload of formal health workers [48,49,50].

South Sudan faces some of the most severe health-workforce challenges globally due to conflict and limited infrastructure. The health workforce is small, with most of the health facilities functioning as a result of donor funding [51]. Frequent insecurity disrupts service delivery and deters workers from remaining in the profession [52]. Despite these challenges, there is growing international support to train health workers locally and build capacity for a resilient health system. Data was collected from five states and counties—Warrap (Twic East, Tonj South), Western Bahr el Ghazal (Wau), Eastern (Kapoeta South), Central Equatoria (Juba and Kajokeji), and Western Equatoria (Yambio, Maridi, and Nzara).

In Senegal, data was collected from five districts: Kolda, Sédhiou, Goudomp, Matam, and Guédiawaye. The expanding health infrastructure in Senegal has been accompanied by a growing health workforce, which is attributed to reinforced training programs that built on the national training plan for health personnel that was established in 1996 [53]. The country has a strong tradition of community-based healthcare, with health posts and community health workers playing vital roles. However, disparities in workforce distribution and insufficient financial incentives remain significant barriers to achieving equitable health-service delivery. Recent investments in digital health have improved workforce efficiency and data management, though these initiatives are still in the early stages [54,55].

### 2.3. Study Respondents

The health-workforce data was collected from health facilities and community health workers present in the health facilities at the time of the survey. The study included healthcare workers who had worked in primary health facilities (health centers, dispensaries, health posts, and nursing homes) for more than one year. The facility in-charge and 2 healthcare workers per facility, i.e., nurses/midwives, the Clinical Officer, Medical Officer and admins, were interviewed. At the community level, Community Health Volunteers working within the catchment area of the facility targeted were interviewed.

### 2.4. Data Collection

The job-satisfaction assessment tool was adopted from a validated tool by Alpern et.al (2013) [56], originally developed for use in low-income healthcare settings. The tool was reviewed by staff from all the country offices to ensure contextual relevance and organization alignment. The survey data collection tool was organized into five dimensions: employee–employer relationships, remuneration and recognition, professional development, physical environment, and supportive supervision, to effectively assess job satisfaction and health among workers. The responses were measured on a 5-point Likert scale of 0: Not applicable, 1: Very dissatisfied, 2: Dissatisfied, 3: Neutral, 4: Satisfied, and 5: Very satisfied. Tools were collaboratively reviewed by countries and aligned to fit the context, and questions for each dimension were drafted from the existing literature and practice to ensure validity. The main tool in the English version was then translated to Kiswahili for Kenya and Tanzania, Arabic for South Sudan, Amharic for Ethiopia, French for Senegal, and Chichewa for Zambia and Malawi. Each country conducted standardized training sessions for research assistants and embarked on a pretest of tools to ensure reliability. Teams converged after the pretest exercise to discuss revisions required based on the pretest; updates were incorporated into the tools, including KoboCollect forms, before the actual data collection. Data collection continued simultaneously from November 2023 to January 2024 in all the countries using the KoboCollect tool, and the data was stored in secure Kobo servers. The data was then downloaded into Excel workbooks for cleaning and later uploaded into R version 4.2.3 (c) 2023, The R Foundation for Statistical Computing, for analysis. The satisfaction survey tool for health workers has been added as a Appendix A.

### 2.5. Sample-Size Determination

Each of the countries used a two-stage cluster sampling formula to compute the number of health facilities and health workers per facility to be interviewed. This method ensured a representative sample of health workers across diverse geographic and facility contexts, while accounting for the hierarchical structure of health systems.

The sample sizes were calculated using the formulae where the sample size (n) was calculated based on a 95% confidence interval (Z = 1.96) and a margin of error (MOE) of 0.05, adjusted for clustering using the design effect (DEFF). The DEFF accounted for variability at two levels: primary and secondary clusters. It is calculated as DEFF = 1 + (M1 − 1) × ICC1 + (M2ij − 1) × ICC2ij, where M1 was the average size of primary clusters (number of health workers per subcounty per 10,000 population), M2ij was the average number of health workers in a health facility (HF) within a subcounty, ICC1 was the intra-cluster correlation at the primary cluster level, and ICC2ij the intra-cluster correlation at the secondary cluster level. Each country adjusted its calculated sample size by 10% to account for potential non-responses.$$n=\left[\frac{Z^{2}\cdot \pi_{1}\cdot \pi_{2ij}\cdot M_{1}\cdot M_{2ij}}{MOE^{2}\cdot DEFF}\right]$$

The study employed a two-stage cluster sampling design to select health facilities and health workers in each of the eight countries. This approach aimed to ensure a representative sample of health workers across various regions while accounting for the hierarchical structure of the data. The sampling procedure can be broken down as follows.

In the first stage, a list of eligible health facilities was compiled from each country’s health-department database. The facilities were selected using a probability proportional to size (PPS) method to ensure that facilities with larger populations of health workers had a higher chance of being selected. This approach was applied to all health facilities within the defined geographic regions of each country, considering factors like urban vs. rural settings and facility type (e.g., hospitals, health centers).

In the second stage, within each selected health facility, purposive sampling was utilized to select health-facility incharges. Additionally, two health workers present at the facility during the day of the interview were randomly selected. However, in some cases, the total number of respondents per facility exceeded the straightforward ratio of 1 facility in-charge and 2 staff members. This discrepancy was due to purposeful oversampling to account for potential non-responses and to ensure sufficient representation of health workers across different cadres and demographics, enhancing the statistical robustness and generalizability of the findings. If some health workers were unavailable for interviews, additional staff were selected to maintain the desired sample size.

The health-worker sample was distributed equally across health-facilities samples, to obtain a probable number of persons to interview per facility. In smaller facilities with fewer staff than the required number, all available workers were interviewed. In contrast, larger facilities could support additional respondents, ensuring the sample targets were met. All of the targeted respondents participated in the survey, resulting in a 100% response rate. This comprehensive participation strengthened the reliability of the data and allowed for richer analysis of the factors influencing job satisfaction among primary healthcare workers.

Table 1 (referenced below) presents the number of health facilities and health workers interviewed in each country. The distribution of health-worker interviews across facilities was proportionally allocated to ensure representativeness.

### 2.6. Data Analysis 

For analysis, the quantitative data underwent univariate (descriptive) and bivariate (cross-tabulation) analysis, with the results presented in tables disaggregated by country, gender, and age categories. The data was grouped into five themes that spoke about different elements that make up job satisfaction. These themes were analyzed separately, but their results collectively formed overall job satisfaction. These were

Employer–Employee RelationshipThis component of job satisfaction was assessed by looking at factors such as the support received from the management, the evaluation of work based on a fair system of performance standards, and the extent to which the institutional rules in place made it easy to work;RemunerationWhen evaluating remuneration as an aspect of job satisfaction, different components were considered. These included the pay being commensurate with the amount of work performed, the pay being commensurate with one’s skills, the bonuses and allowances received, and whether the worker’s efforts were being recognized;Professional DevelopmentJob satisfaction as it relates to the professional development of health workers was examined by looking at the Training Opportunities for Professional Development that were available to the health workers, their satisfaction with the Space and Opportunities to Learn New Skills, and the Quality of Blended Training Received;Physical EnvironmentThis was composed of factors like the Protection against Occupational Hazards, the Sufficiency of Work Equipment, the Safety of the Physical Environment, and satisfaction regarding the housing facilities/housing allowance provided;Supportive SupervisionSupportive supervision was analyzed by looking at the Coaching Received from the Direct Supervisor, the extent to which suggestions are heard by the supervisor, and the availability of the supervisor to answer work-related questions

## 3. Results

### 3.1. Demographic and Professional Characteristics

A total of 1711 health workers participated in the study, with representation from Tanzania (36%), Uganda (18%), Senegal (11%), Ethiopia (10%), Kenya (8%), Malawi (7%), Zambia (5%), and South Sudan (4%). The median length of service among the respondents was 8 years, with Zambia reporting the longest median at 11 years and Ethiopia the shortest at 7 years. Overall, the median facility service length was 4 years, with Zambia and Senegal having the highest at 6 years and Ethiopia and Malawi having the lowest at 3 years. Gender distribution revealed that women made up the majority of health workers in most countries, apart from Malawi and South Sudan. Overall, 59% of respondents were female, with Zambia exhibiting the highest proportion (72%), followed by Tanzania (64%), Senegal (60%), Kenya (59%), Uganda (58%), Ethiopia (55%), South Sudan (45%), and Malawi (43%). The median age of respondents was 36 years, with Zambia reporting the highest median at 42 years and Ethiopia reporting the lowest at 30 years. Community-based health workers made up 40% of the workforce, with the highest representation in Senegal (77%) and the lowest in Ethiopia (8.2%), while facility-based health workers accounted for the remaining 60%. See Table 2 for details.

### 3.2. Job Satisfaction

Satisfaction with the employer–employee relationship was highest in Zambia (80%) and lowest in Tanzania (16%). Remuneration satisfaction was notably higher in Senegal (63%) and was followed by Zambia (49%), but was extremely low in Malawi (9.8%) and Ethiopia (2.3%). Overall, 44% of the respondents reported being satisfied with their professional development, with Uganda leading (62%) and Ethiopia reporting the lowest satisfaction (29%). Satisfaction with the physical environment stood at 27% overall, with Uganda reporting the highest (40%) and Kenya the lowest (12%). Satisfaction with supervisory support was at 62%, with Zambia showing the highest satisfaction levels (73%) and Ethiopia the lowest (30%). See Table 3 below.

### 3.3. Predictors of Employer–Employee Relationship Satisfaction

The multivariable analysis of employer–employee-relationship satisfaction revealed significant variations based on country, demographics, and job-related factors (Table 4). Health workers in Zambia showed the highest odds of satisfaction with their employer (OR = 4.97, 95% CI: 2.48–10.30, *p* < 0.001), followed by Uganda (OR = 2.17, 95% CI: 1.35–3.49, *p* = 0.001). Conversely, Tanzania struggles the most (OR = 0.17, 95% CI: 0.11–0.27, *p* < 0.001), indicating widespread dissatisfaction. Satisfaction was also associated with longer service duration. Health workers satisfied with their employer had a mean service duration of 11.1 years, compared to 9.7 years for those dissatisfied (OR = 1.03, 95% CI: 1.01–1.06, *p* = 0.014). Professional cadre influenced satisfaction, with facility-based workers being more likely to report satisfaction than community-based workers (OR = 3.49, 95% CI: 2.51–4.89, *p* < 0.001). Job attributes such as pay, professional growth opportunities, the physical environment, and supervisory support were strongly correlated with employer satisfaction. Workers satisfied with their pay were 74% more likely to be satisfied with their employer (OR = 1.74, 95% CI: 1.25–2.41, *p* = 0.001). Positive perceptions of professional development (OR = 2.24, 95% CI: 1.73–2.92, *p* < 0.001) and the workplace environment (OR = 1.98, 95% CI: 1.47–2.66, *p* < 0.001) significantly increased satisfaction. The strongest association was with supervisory support, where workers who felt supported were over three times more likely to be satisfied with their employer (OR = 3.34, 95% CI: 2.51–4.45, *p* < 0.001).

### 3.4. Predictors of Remuneration Satisfaction

Table 5 presents the predictors of remuneration satisfaction across the eight countries. Health workers in Senegal were the most satisfied with their pay (OR = 26.34, 95% CI: 10.04–90.98, *p* < 0.001), followed by those in Zambia (OR = 19.47, 95% CI: 6.98–69.82, *p* < 0.001). In contrast, only 2.3% of Ethiopian health workers reported satisfaction with their remuneration. Significant satisfaction levels were also observed in Kenya (OR = 8.49, 95% CI: 3.10–30.02, *p* < 0.001), Uganda (OR = 8.86, 95% CI: 3.46–30.13, *p* < 0.001), and Tanzania (OR = 4.98, 95% CI: 1.94–16.98, *p* = 0.003). While univariable analysis suggested an association between length of service and remuneration satisfaction (OR = 1.05, 95% CI: 1.03–1.07, *p* < 0.001), this was not significant in the multivariable model (OR = 1.02, 95% CI: 0.99–1.05, *p* = 0.281). Facility-based health workers were significantly less likely to be satisfied with their pay than community-based workers (OR = 0.20, 95% CI: 0.15–0.28, *p* < 0.001). Job attributes also played a critical role in remuneration satisfaction. Positive employer–employee relationships (OR = 1.64, 95% CI: 1.17–2.31, *p* = 0.004), professional-development opportunities (OR = 1.59, 95% CI: 1.18–2.15, *p* = 0.002), and a favorable physical environment (OR = 1.57, 95% CI: 1.16–2.12, *p* = 0.003) were all associated with higher remuneration satisfaction. However, supervisory support was not a significant predictor in the multivariable analysis (OR = 1.11, 95% CI: 0.80–1.56, *p* = 0.525).

### 3.5. Predictors of Health-Worker Satisfaction with Professional Development

Table 6 outlines the predictors of health-worker satisfaction with professional development. Health workers satisfied with employer–employee relationships were significantly more likely to be satisfied with their professional development (OR = 2.20, 95% CI: 1.69–2.85, *p* < 0.001). Satisfaction with remuneration (OR = 1.59, 95% CI: 1.19–2.13, *p* = 0.002), the physical environment (OR = 1.71, 95% CI: 1.32–2.22, *p* < 0.001), and supervisory support (OR = 3.58, 95% CI: 2.78–4.62, *p* < 0.001) were also strong predictors. While satisfaction varied by country, it was not a significant predictor after adjusting for other factors. Although health workers in Uganda and South Sudan reported higher satisfaction, these findings were not statistically significant in the multivariable analysis. Demographic factors such as sex, age, and professional cadre had no significant impact, nor did facility tenure or total years of service.

### 3.6. Predictors of Physical-Environment Satisfaction

The predictors of physical-environment satisfaction are presented in Table 7. Satisfaction with the physical environment was significantly influenced by employer–employee relationships (OR = 1.96, 95% CI: 1.46–2.64, *p* < 0.001), remuneration satisfaction (OR = 1.59, 95% CI: 1.18–2.14, *p* = 0.002), professional-development satisfaction (OR = 1.68, 95% CI: 1.29–2.19, *p* < 0.001), and supervisory support (OR = 2.14, 95% CI: 1.59–2.91, *p* < 0.001). Healthcare workers in Kenya (OR = 0.26, 95% CI: 0.13–0.53, *p* < 0.001) and Zambia (OR = 0.49, 95% CI: 0.24–0.99, *p* = 0.046) were significantly less likely to report satisfaction with their physical environment compared to Ethiopian workers.

Interestingly, while the length of service did not emerge as a significant predictor, facility-based workers were significantly less likely to report satisfaction with the physical environment compared to community-based workers (OR = 0.31, 95% CI: 0.23–0.41, *p* < 0.001).

### 3.7. Predictors of Supportive Supervision Satisfaction

The predictors of satisfaction with supportive supervision are summarized in Table 8. Satisfaction was most strongly associated with remuneration (OR = 2.39, 95% CI: 1.77–3.22, *p* < 0.001) and employer–employee relationships (OR = 2.51, 95% CI: 1.90–3.30, *p* < 0.001). Similarly, satisfaction with professional development (OR = 1.95, 95% CI: 1.41–2.68, *p* < 0.001) and the physical environment (OR = 2.10, 95% CI: 1.59–2.76, *p* < 0.001) were significant predictors of supportive supervision satisfaction. Tanzania (OR = 6.53, 95% CI: 4.19–10.33, *p* < 0.001) and Kenya (OR= 5.02, 95% CI: 2.88–8.89, *p* < 0.001) showed the highest and most significant levels of satisfaction with supervision, while health workers in Malawi (OR =1.86, 95% CI: 1.05–3.29, *p* = 0.032) were significantly less likely to report satisfaction.

## 4. Discussion

This study offers insights into the demographic and professional characteristics of health workers across eight African countries, highlighting variations in workforce composition, tenure, and gender distribution. Tanzania had the largest representation (36%), followed by Uganda (18%) and Senegal (11%), while Zambia (5%) and South Sudan (4%) had the smallest, reflecting differences in workforce distribution and study participation. The median length of service was 8 years, with Zambia reporting the longest at 11 years and Ethiopia the shortest at 7 years. Zambia and Senegal had the longest median facility-service duration (6 years), suggesting a relatively stable workforce, whereas Ethiopia and Malawi had the shortest (3 years), potentially indicating higher turnover or workforce mobility.

The average length of service for healthcare workers in Sub-Saharan Africa (SSA) is significantly shorter compared to high-income regions, primarily due to high attrition rates driven by migration, burnout, and poor working conditions. Studies suggest that many doctors and nurses leave their positions within 5–10 years, with some departing even sooner for better opportunities abroad [57,58,59]. For instance, the World Health Organisation (WHO) reports that 55% of African-trained doctors emigrate to wealthier nations, particularly to Europe, North America, and the Gulf states, within a few years of qualification [60]. Nurses and midwives tend to stay slightly longer, averaging 5–10 years, but these roles still face high turnover due to low wages and heavy workloads [57,61,62]. Community health workers (CHWs), who play a crucial role in rural healthcare, often have even shorter tenures (2–5 years) because of informal employment structures and inadequate compensation [63].

The high turnover in some countries could point to job dissatisfaction, poor working conditions, or a lack of career-growth opportunities. Countries such as Zambia and Uganda, which showed longer service duration, may do so because of better retention policies, but this could also indicate fewer opportunities for mobility or promotion. While the findings suggest that health workers have multiple years of service, sub-Saharan Africa is faced with challenges such as corruption, poor working conditions, and a sub-optimal use of time, which end up affecting overall healthcare delivery [64].

Women comprised the majority of health workers in all countries except Malawi and South Sudan. Overall, 59% of respondents were female, with the highest proportion in Zambia (72%), whereas Malawi (43%) and South Sudan (45%) have a more balanced or male-dominated workforce. Such variations may have implications for gender-sensitive health services, particularly in maternal and child-health programs, where female health workers often play a crucial role. The predominance of female health workers aligns with global trends, particularly in nursing and community health roles where women make up 70% of the workforce [65].

The median age of health workers was 36 years, with Zambia having the oldest median at 42 years and Ethiopia the youngest at 30 years. The relatively older workforce in Zambia suggests a more experienced health sector, whereas Ethiopia’s younger workforce could indicate recent recruitment efforts or a higher attrition of older health professionals. However, Africa’s general population is notably young, with a median age of about 20 years. This youthful demographic suggests that the health workforce may also be relatively young, which calls for in-service training, mentorship, and career-progression opportunities to improve care [66].

This study assessed job satisfaction among 1711 health workers across eight sub-Saharan African countries, Ethiopia, Kenya, Malawi, Senegal, South Sudan, Tanzania, Uganda, and Zambia, focusing on five dimensions: employer–employee relationships, remuneration and recognition, professional development, the physical work environment, and supportive supervision. The cross-country differences in health-worker job satisfaction observed in this study reflect broader structural, policy, and contextual factors that influence health systems across sub-Saharan Africa. For instance, Zambia’s relatively high satisfaction levels across multiple domains, including employer–employee relationships and supportive supervision, may be attributed to its ongoing investments in health-workforce governance and human-resource management. Studies have shown that the implementation of supportive supervision frameworks and performance-management systems in Zambia has contributed to improved health-worker morale and retention [67,68]. In contrast, Tanzania and Ethiopia exhibited some of the lowest satisfaction levels, particularly in employer–employee relations and remuneration. Previous studies in Tanzania have highlighted systemic issues such as unclear promotion structures, delayed salaries, and insufficient recognition, which diminish health-worker motivation [69]. Similarly, in Ethiopia, chronic underinvestment in human resources and limited professional-development opportunities in rural areas have been linked to dissatisfaction and workforce attrition [70,71].

These findings align with previous research highlighting the critical role of remuneration, professional development, and working conditions in health-worker satisfaction. Studies in Uganda and Zambia identified employer–employee relationships and supervisory support as key satisfaction drivers [72]. Tanzania reported significantly lower satisfaction at 16%, indicating potential challenges in employer engagement, communication, or workplace policies [13]. A similar study in Kenya, Uganda, and Zambia identified employer–employee relationships and supervisory support as key satisfaction drivers [73]. Positive employer–employee relationships foster a sense of workplace belonging and trust, which directly influences satisfaction with the physical work environment [74], leading to a more positive supervisory experience [16]. Additionally, the existing literature emphasizes the importance of supportive work environments, fair compensation, and effective supervision in enhancing job satisfaction among healthcare professionals [30].

The high satisfaction with remuneration observed in Senegal aligns with findings from evaluations of its performance-based financing (PBF) programs, which have introduced financial incentives and improved accountability in health facilities [75]. The low satisfaction with physical work environments in Kenya contrasts with its relatively well-resourced urban health facilities. This, however, suggests persistent inequalities in health infrastructure in urban and rural areas, as reported in studies examining health-worker experiences in county-level facilities [76]. Job attributes such as pay, professional-growth opportunities, the physical environment, and supervisory support were strongly correlated with employer satisfaction. This corroborates findings from a similar study conducted in Ethiopia, which reported salary and incentives, benefit packages, recognition by management, patient appreciation, the working environment, developmental opportunities, better management, clear communication, and staff working relationships as strong predictors of job satisfaction [77].

Health-worker satisfaction with remuneration in Africa varies across regions and is influenced by multiple factors. A study conducted in Ethiopia documented that despite governmental efforts to enhance health infrastructure and workforce numbers, national health services often struggle to attract and retain health workers. This challenge is partly due to inadequate remuneration and insufficient attention to incentives and motivation, leading to decreased productivity and increased turnover among health professionals [77]. Facility-based health workers were significantly less satisfied with their remuneration compared to community-based workers. This could be due to differences in workload, compensation models, or perceived fairness in pay structures, such as working conditions, resource availability, and management practices. The findings are similar to a study [78] which consistently identified poor remuneration, limited career growth, and inadequate working conditions as major drivers of dissatisfaction among health workers in low-resource settings. These results align with previous studies emphasizing the role of financial incentives and workplace relationships in enhancing supervisory satisfaction [30]. Additionally, a study in Uganda and Zambia has demonstrated that satisfaction with pay and management quality significantly influences job satisfaction and retention [79].

Uganda’s lead in professional-development satisfaction is supported by evidence of targeted training programs and continuous professional education supported by the government and partners [75]. This aligns with findings from Namazzi who documented increased training coverage and mentoring in maternal and child-health programs. Conversely, Malawi’s low scores on remuneration and recognition reflect constraints associated with wage freezes and heavy reliance on donor-funded vertical programs, which often bypass national HRH priorities [80].

These cross-country comparisons emphasize that while some countries have adopted effective workforce-strengthening strategies, others face persistent gaps in management, supervision, and support systems. These differences underscore the need for tailored health workforce policies that are responsive to national contexts, as also emphasized in the WHO’s Global Strategy on Human Resources for Health [33]. Job satisfaction levels among health workers varied significantly across the eight countries studied, reflecting the influence of broader systemic, economic, and institutional differences. The higher satisfaction scores in Zambia and Senegal may reflect relatively better remuneration systems, stronger supervisory structures, or more coordinated health-workforce strategies, while lower satisfaction in Ethiopia and Malawi may stem from chronic health-system underfunding, limited professional growth, and inadequate infrastructure—patterns consistent with previous findings in low-resource settings [30,81,82]. These disparities mirror structural health-system differences that are also evident when comparing sub-Saharan Africa to high-income countries. For instance, in many OECD countries, job satisfaction among health workers tends to be higher due to better salaries, stronger professional-development pathways, safer working environments, and more participatory management practices [83,84].

In contrast to sub-Saharan Africa, where dissatisfaction is often rooted in resource scarcity and systemic weaknesses, studies from countries such as Sweden, the Netherlands, and Canada point to job satisfaction being influenced more by workload balance, autonomy, and opportunities for innovation [35,85]. Moreover, the presence of robust health-worker unions and stronger policy mechanisms for staff well-being and accountability in high-income countries contributes to greater professional agency and morale. These comparative insights highlight the importance of tailoring health-workforce strategies to local contexts while learning from successful models of governance, training, and incentives employed in more developed systems.

Organizations that prioritize career development contribute positively to health workers’ satisfaction, as documented by Essex County Council in the UK, where emphasis on the career progression of its social workers led to improved morale and retention [86]. Similarly, a meta-analysis in Ethiopia identified salary, recognition, professional development, and supportive supervision as strong predictors of job satisfaction. The observed country-specific variations underscore the necessity for tailored interventions, as highlighted in studies comparing job satisfaction across different African countries.

Although most studies conducted focus on specific countries, this study, however, adds to the evidence by providing multi-country comparative data using a standardized methodology, allowing for more nuanced cross-national analysis within the sub-Saharan African context. Strengths of the study include its large, multi-country sample (n = 1711), use of a standardized data-collection tool, and a robust sampling methodology, which increase the generalizability and reliability of findings across diverse health systems. The study also assessed multiple facets of job satisfaction through multivariable regression analysis, which provided valuable insights into predictors of job satisfaction, providing a holistic view of job satisfaction in SSA countries.

Limitations of the study include potential response bias due to self-reported data and variability in the interpretation of satisfaction scales. Furthermore, although efforts were made to include a representative sample, oversampling in some facilities and reliance on availability during data collection may have introduced selection bias. The cross-sectional design also limits causal inferences, and differences in healthcare systems across countries may affect comparability [72]. Lastly, the study did not account for the impact of external factors like political instability or economic conditions, which could influence job satisfaction. The cross-sectional design restricts causal inference between predictors and satisfaction. Self-reported responses may be influenced by social desirability or cultural biases in interpreting Likert-scale items [87,88]. Furthermore, the study did not account for facility-level or regional health-system factors, such as staffing ratios, management style, or workload, which may independently affect satisfaction [89]. Future research using longitudinal, mixed-methods, or multilevel designs could provide more nuanced and generalizable evidence to inform reforms in health-worker motivation and retention.

The study provides critical, actionable insights for health-workforce strengthening in sub-Saharan Africa, grounded in Herzberg’s Two-Factor Theory of Motivation. According to the theory, job satisfaction is influenced by two categories of factors: motivators (e.g., recognition, professional development, meaningful work) and hygiene factors (e.g., pay, supervision, working conditions). This study demonstrates that both sets of factors are significantly associated with health-worker satisfaction.

Key findings—such as the strong association between supervisory support, remuneration, employer–employee relationships, and job satisfaction—highlight the need for strategic investment in both intrinsic motivators and extrinsic conditions. In particular, supportive supervision emerged as the most powerful predictor of satisfaction (OR = 3.34), underscoring the importance of strengthening leadership and accountability mechanisms within health systems.

The study provides critical evidence to guide health-workforce interventions in sub-Saharan Africa. Policymakers and health-system planners should prioritize improving employer–employee relationships, investing in fair and transparent remuneration systems, enhancing supervision structures, and upgrading physical work environments. The country-specific variations underscore the need for localized strategies rather than generic approaches. Improving employer–employee relationships, ensuring fair remuneration, enhancing professional development opportunities, and providing supportive supervision are vital for boosting health-worker satisfaction and retention. These findings can serve as a baseline for evaluating the effectiveness of workforce reforms and informing donor investments in health-workforce strengthening.

## 5. Conclusions

This study provides compelling evidence that job satisfaction among health workers in sub-Saharan Africa is shaped by a complex interplay of factors, including employer–employee relationships, remuneration, professional-development opportunities, physical working conditions, and supportive supervision. Significant cross-country variations were observed, underscoring the critical need for context-specific workforce strategies. Countries such as Zambia and Senegal demonstrated comparatively higher levels of satisfaction across several domains, including remuneration and supervisory support, whereas Ethiopia and Tanzania consistently reported the lowest satisfaction, particularly with working conditions and managerial relationships.

The findings reinforce the importance of Herzberg’s Two-Factor Theory, where both hygiene factors (e.g., pay, work environment) and motivators (e.g., recognition, advancement) must be addressed to enhance satisfaction and reduce turnover. The strong association between supervisory support and overall satisfaction highlights the value of effective management practices in motivating and retaining staff.

Moreover, the study reveals meaningful workforce dynamics, including gender imbalances, varied lengths of service, and differing distributions between facility- and community-based workers, which can influence satisfaction and performance. For instance, Zambia’s longer service duration may point to better retention practices, while Ethiopia’s lower tenure suggests a need to investigate and address drivers of early attrition.

To strengthen health systems across the region, targeted interventions should focus on building supportive supervision structures, ensuring fair and transparent compensation systems, fostering professional-growth opportunities, and improving physical working environments. Given the diversity across countries, one-size-fits-all solutions are unlikely to succeed. Policymakers and development partners must prioritize tailored approaches that reflect the unique health system contexts, labor-market dynamics, and demographic realities of each country.

Finally, the study serves as an important baseline for measuring progress and guiding health workforce reform. It offers actionable evidence for governments, donors, and implementers aiming to build a resilient, motivated health workforce—one that is essential for delivering quality care, reducing health inequities, and achieving broader health and development goals in sub-Saharan Africa.

## Figures and Tables

**Figure 1 ijerph-22-01108-f001:**
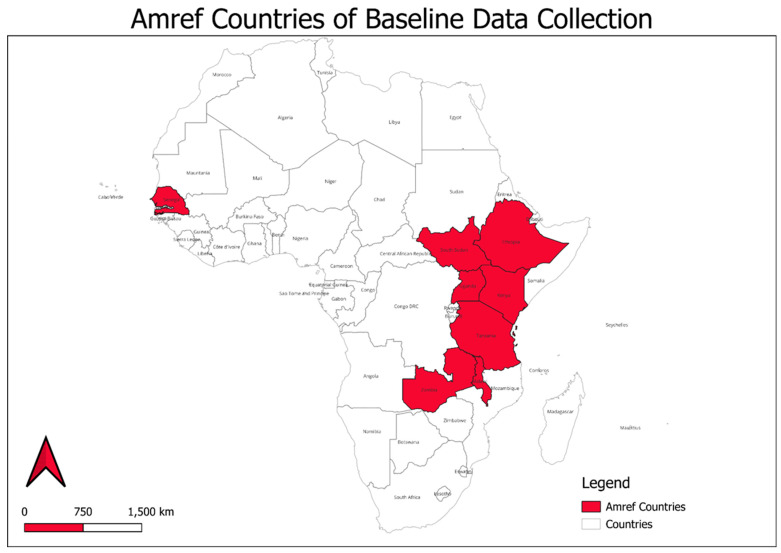
Amref Health Africa: implementing countries. Created using QGIS 3.34.

**Table 1 ijerph-22-01108-t001:** Sample size by country.

Sample Type	Ethiopia	Kenya	Tanzania	Malawi	South Sudan	Uganda	Senegal	Zambia	Total
Respondents	171	143	615	123	66	310	192	91	1711
Response Rate	100%	100%	100%	100%	100%	100%	100%	100%	100%
% proportion of sample size against the total health workforce in the area of study	9.99%	8.36%	35.94%	7.19%	3.86%	18.12%	11.22%	5.32%	100.00%
Health Facilities	65	49	93	32	64	21	13	20	357

**Table 2 ijerph-22-01108-t002:** Characteristics of the health workforce.

Characteristic	Overall (n = 1711)	Ethiopia (n = 171.10%)	Kenya (n = 143.8%)	Malawi (n = 123.7%)	Senegal (n= 192.11%)	South Sudan (n = 66.4%)	Tanzania (n = 615.36%)	Uganda (n = 310.18%)	Zambia (n = 91.5%)
Length of Service (Years)									
Median (lower, upper years)	8 (4, 14)	7 (4, 10)	10 (6, 14)	8 (3, 16)	8 (5, 13)	8 (4, 14)	8 (3, 13)	10 (5, 16)	11 (5, 18)
Service Length in Facility (Years)									
Median (lower, upper years)	4 (2, 9)	3 (2, 5)	4 (2, 7)	3 (1, 6)	6 (3, 11)	4 (2, 6)	5 (2, 9)	5 (2, 13)	6 (4, 13)
Respondent Sex									
Female	1003 (59%)	93 (55%)	83 (59%)	52 (43%)	113 (60%)	29 (45%)	391 (64%)	177 (58%)	65 (72%)
Male	688 (41%)	77 (45%)	57 (41%)	69 (57%)	76 (40%)	36 (55%)	220 (36%)	128 (42%)	25 (28%)
Respondent Age (Years)									
Median (lower, upper years)	36 (30, 45)	30 (28, 35)	37 (32, 43)	37 (30, 45)	36 (30, 44)	35 (31, 46)	36 (30, 47)	37 (32, 47)	42 (32, 54)
Cadre									
Community-Based	684 (40%)	14 (8.2%)	42 (29%)	27 (22%)	140 (77%)	11 (17%)	254 (41%)	149 (49%)	47 (52%)
Facility-Based	1005 (60%)	156 (92%)	101 (71%)	95 (78%)	43 (23%)	52 (83%)	361 (59%)	153 (51%)	44 (48%)

**Table 3 ijerph-22-01108-t003:** Job satisfaction of health workforce.

Characteristic	Overall (n = 1711)	Ethiopia (n = 171.10%)	Kenya (n = 143.8%)	Malawi (n = 123.7%)	Senegal (n = 192.11%)	South Sudan (n = 66.4%)	Tanzania (n = 615.36%)	Uganda (n = 310.18%)	Zambia (n = 91.5%)
Employee–Employer Satisfaction									
Not Satisfactory	1004 (59%)	104 (61%)	57 (40%)	71 (58%)	107 (56%)	32 (48%)	510 (84%)	105 (34%)	18 (20%)
Satisfactory	693 (41%)	67 (39%)	85 (60%)	52 (42%)	83 (44%)	34 (52%)	94 (16%)	205 (66%)	73 (80%)
Remuneration Satisfaction									
Not Satisfactory	1258 (74%)	167 (98%)	107 (75%)	111 (90%)	71 (37%)	55 (83%)	505 (82%)	196 (63%)	46 (51%)
Satisfactory	453 (26%)	4 (2.3%)	36 (25%)	12 (9.8%)	121 (63%)	11 (17%)	110 (18%)	114 (37%)	45 (49%)
Professional-Development Satisfaction									
Not Satisfactory	929 (56%)	121 (71%)	87 (61%)	74 (62%)	104 (56%)	30 (46%)	358 (61%)	117 (38%)	38 (42%)
Satisfactory	742 (44%)	49 (29%)	55 (39%)	46 (38%)	81 (44%)	35 (54%)	231 (39%)	193 (62%)	52 (58%)
Physical-Environment Satisfaction									
Not Satisfactory	1232 (73%)	141 (82%)	125 (88%)	97 (79%)	120 (66%)	46 (71%)	456 (74%)	185 (60%)	62 (68%)
Satisfactory	467 (27%)	30 (18%)	17 (12%)	26 (21%)	63 (34%)	19 (29%)	159 (26%)	124 (40%)	29 (32%)
Support Supervisory Satisfaction									
Not Satisfactory	647 (38%)	120 (70%)	43 (30%)	66 (54%)	75 (40%)	30 (45%)	195 (32%)	93 (30%)	25 (27%)
Satisfactory	1058 (62%)	51 (30%)	98 (70%)	56 (46%)	114 (60%)	36 (55%)	420 (68%)	217 (70%)	66 (73%)

**Table 4 ijerph-22-01108-t004:** Predictors of employer–employee-relationship satisfaction among health workforce in 8 African countries.

Health Workforce: Characteristics		OR (Univariable)	OR (Multivariable)
Country Name	Ethiopia	-	-
	Kenya	2.31 (1.47–3.66, *p* < 0.001)	1.60 (0.94–2.72, *p* = 0.085)
	Malawi	1.14 (0.71–1.82, *p* = 0.594)	0.99 (0.57–1.69, *p* = 0.957)
	Senegal	1.20 (0.79–1.84, *p* = 0.386)	1.05 (0.60–1.82, *p* = 0.871)
	South Sudan	1.65 (0.93–2.93, *p* = 0.087)	1.18 (0.60–2.33, *p* = 0.632)
	Tanzania	0.29 (0.20–0.42, *p* < 0.001)	0.17 (0.11–0.27, *p* < 0.001)
	Uganda	3.03 (2.06–4.48, *p* < 0.001)	2.17 (1.35–3.49, *p* = 0.001)
	Zambia	6.30 (3.52–11.74, *p* < 0.001)	4.97 (2.48–10.30, *p* < 0.001)
Length of Service (Years)	Mean (SD)	1.02 (1.01–1.03, *p* = 0.001)	1.03 (1.01–1.06, *p* = 0.014)
Service Length in Facility (Years)	Mean (SD)	1.01 (1.00–1.03, *p* = 0.055)	0.99 (0.96–1.02, *p* = 0.469)
Respondent Sex	Female	-	-
	Male	1.03 (0.84–1.26, *p* = 0.772)	1.09 (0.84–1.40, *p* = 0.523)
Respondent Age	Mean (SD)	1.01 (1.00–1.02, *p* = 0.085)	0.99 (0.97–1.01, *p* = 0.314)
Cadre	Community Based	-	-
	Facility Based	1.37 (1.12–1.67, *p* = 0.002)	3.49 (2.51–4.89, *p* < 0.001)
Renumeration Satisfaction	not satisfactory	-	-
	satisfactory	1.77 (1.42–2.20, *p* < 0.001)	1.74 (1.25–2.41, *p* = 0.001)
Professional Development Satisfaction	not satisfactory	-	-
	satisfactory	3.12 (2.55–3.83, *p* < 0.001)	2.24 (1.73–2.92, *p* < 0.001)
Physical Environment Satisfaction	not satisfactory	-	-
	satisfactory	2.03 (1.63–2.52, *p* < 0.001)	1.98 (1.47–2.66, *p* < 0.001)
Support Supervisory Satisfaction	not satisfactory	-	-
	satisfactory	3.22 (2.59–4.00, *p* < 0.001)	3.34 (2.51–4.45, *p* < 0.001)

OR = Odds Ratio, SD = Standard Deviation.

**Table 5 ijerph-22-01108-t005:** Predictors of renumeration satisfaction among health workforce in 8 African countries.

Health Workforce: Renumeration		OR (Univariable)	OR (Multivariable)
Country Name	Ethiopia	-	-
	Kenya	14.05 (5.43–47.95, *p* < 0.001)	8.49 (3.10–30.02, *p* < 0.001)
	Malawi	4.51 (1.53–16.47, *p* = 0.011)	3.43 (1.12–12.88, *p* = 0.043)
	Senegal	71.15 (28.55–238.32, *p* < 0.001)	26.34 (10.04–90.98, *p* < 0.001)
	South Sudan	8.35 (2.73–31.11, *p* < 0.001)	2.72 (0.71–11.50, *p* = 0.149)
	Tanzania	9.09 (3.75–29.99, *p* < 0.001)	4.98 (1.94–16.98, *p* = 0.003)
	Uganda	24.28 (9.94–80.39, *p* < 0.001)	8.86 (3.46–30.13, *p* < 0.001)
	Zambia	40.84 (15.59–140.80, *p* < 0.001)	19.47 (6.98–69.82, *p* < 0.001)
Length of Service (Years)	Mean (SD)	1.02 (1.01–1.03, *p* = 0.005)	1.00 (0.97–1.03, *p* = 0.880)
Service Length in Facility (Years)	Mean (SD)	1.05 (1.03–1.07, *p* < 0.001)	1.02 (0.99–1.05, *p* = 0.281)
Respondent Sex	Female	-	-
	Male	0.98 (0.78–1.22, *p* = 0.836)	1.17 (0.89–1.54, *p* = 0.268)
Respondent Age	Mean (SD)	1.02 (1.01–1.03, *p* < 0.001)	0.99 (0.97–1.00, *p* = 0.141)
Cadre	Community Based	-	-
	Facility Based	0.15 (0.12–0.20, *p* < 0.001)	0.20 (0.15–0.28, *p* < 0.001)
Employee–Employer Satisfaction	not satisfactory	-	-
	satisfactory	1.77 (1.42–2.20, *p* < 0.001)	1.64 (1.17–2.31, *p* = 0.004)
Professional Development Satisfaction	not satisfactory	-	-
	satisfactory	2.28 (1.83–2.85, *p* < 0.001)	1.59 (1.18–2.15, *p* = 0.002)
Physical Environment Satisfaction	not satisfactory	-	-
	satisfactory	2.65 (2.10–3.33, *p* < 0.001)	1.57 (1.16–2.12, *p* = 0.003)
Support Supervisory Satisfaction	not satisfactory	-	-
	satisfactory	1.87 (1.48–2.37, *p* < 0.001)	1.11 (0.80–1.56, *p* = 0.525)

OR = Odds Ratio, SD = Standard Deviation.

**Table 6 ijerph-22-01108-t006:** Predictors of professional-development satisfaction among health workforce in 8 African countries.

Health Workforce: Professional Development		OR (Univariable)	OR (Multivariable)
Country Name	Ethiopia	-	-
	Kenya	1.56 (0.97–2.51, *p* = 0.065)	0.71 (0.41–1.22, *p* = 0.218)
	Malawi	1.54 (0.93–2.52, *p* = 0.090)	1.11 (0.63–1.94, *p* = 0.716)
	Senegal	1.92 (1.24–3.00, *p* = 0.004)	0.93 (0.54–1.61, *p* = 0.796)
	South Sudan	2.88 (1.60–5.23, *p* < 0.001)	1.97 (0.99–3.93, *p* = 0.053)
	Tanzania	1.59 (1.11–2.32, *p* = 0.014)	0.99 (0.64–1.55, *p* = 0.975)
	Uganda	4.07 (2.74–6.14, *p* < 0.001)	1.52 (0.95–2.46, *p* = 0.085)
	Zambia	3.38 (1.99–5.80, *p* < 0.001)	0.96 (0.51–1.81, *p* = 0.901)
Length of Service (Years)	Mean (SD)	1.02 (1.01–1.03, *p* = 0.001)	1.01 (0.99–1.04, *p* = 0.297)
Service Length in Facility (Years)	Mean (SD)	1.03 (1.01–1.04, *p* < 0.001)	0.98 (0.96–1.01, *p* = 0.146)
Respondent Sex	Female	-	-
	Male	0.95 (0.78–1.16, *p* = 0.621)	0.99 (0.78–1.24, *p* = 0.916)
Respondent Age	Mean (SD)	1.02 (1.01–1.03, *p* < 0.001)	1.01 (0.99–1.03, *p* = 0.254)
Cadre	Community Based	-	-
	Facility Based	0.70 (0.58–0.85, *p* < 0.001)	0.88 (0.66–1.16, *p* = 0.359)
Employee–Employer Satisfaction	not satisfactory	-	-
	satisfactory	3.12 (2.55–3.83, *p* < 0.001)	2.20 (1.69–2.85, *p* < 0.001)
Remuneration Satisfaction	not satisfactory	-	-
	satisfactory	2.28 (1.83–2.85, *p* < 0.001)	1.59 (1.19–2.13, *p* = 0.002)
Physical-Environment Satisfaction	not satisfactory	-	-
	satisfactory	2.69 (2.16–3.36, *p* < 0.001)	1.71 (1.32–2.22, *p* < 0.001)
Support Supervisory Satisfaction	not satisfactory	-	-
	satisfactory	4.63 (3.72–5.80, *p* < 0.001)	3.58 (2.78–4.62, *p* < 0.001)

OR = Odds Ratio, SD = Standard Deviation.

**Table 7 ijerph-22-01108-t007:** Predictors of physical-environment satisfaction among health workforce in 8 African countries.

Health Workforce:		OR (Univariable)	OR (Multivariable)
Country Name	Ethiopia	-	-
	Kenya	0.64 (0.33–1.20, *p* = 0.172)	0.26 (0.13–0.53, *p* < 0.001)
	Malawi	1.26 (0.70–2.26, *p* = 0.439)	0.85 (0.44–1.62, *p* = 0.622)
	Senegal	2.47 (1.51–4.10, *p* < 0.001)	0.61 (0.33–1.11, *p* = 0.100)
	South Sudan	1.94 (0.99–3.76, *p* = 0.050)	1.10 (0.51–2.36, *p* = 0.800)
	Tanzania	1.64 (1.08–2.57, *p* = 0.026)	0.84 (0.51–1.41, *p* = 0.503)
	Uganda	3.15 (2.02–5.03, *p* < 0.001)	0.82 (0.48–1.42, *p* = 0.478)
	Zambia	2.20 (1.22–3.98, *p* = 0.009)	0.49 (0.24–0.99, *p* = 0.046)
Length of Service (Years)	Mean (SD)	1.01 (1.00–1.03, *p* = 0.033)	0.99 (0.96–1.02, *p* = 0.510)
Service Length in Facility (Years)	Mean (SD)	1.04 (1.03–1.06, *p* < 0.001)	1.02 (0.99–1.05, *p* = 0.175)
Respondent Sex	Female	-	-
	Male	0.89 (0.71–1.10, *p* = 0.282)	0.84 (0.65–1.08, *p* = 0.171)
Respondent Age	Mean (SD)	1.02 (1.01–1.03, *p* < 0.001)	0.99 (0.98–1.01, *p* = 0.444)
Cadre	Community Based	-	-
	Facility Based	0.29 (0.23–0.36, *p* < 0.001)	0.31 (0.23–0.41, *p* < 0.001)
Employee–Employer Satisfaction	not satisfactory	-	-
	satisfactory	2.03 (1.63–2.52, *p* < 0.001)	1.96 (1.46–2.64, *p* < 0.001)
Renumeration Satisfaction	not satisfactory	-	-
	satisfactory	2.65 (2.10–3.33, *p* < 0.001)	1.59 (1.18–2.14, *p* = 0.002)
Professional-Development Satisfaction	not satisfactory	-	-
	satisfactory	2.69 (2.16–3.36, *p* < 0.001)	1.68 (1.29–2.19, *p* < 0.001)
Support Supervisory Satisfaction	not satisfactory	-	-
	satisfactory	3.06 (2.39–3.95, *p* < 0.001)	2.14 (1.59–2.91, *p* < 0.001)

OR = Odds Ratio, SD = Standard Deviation.

**Table 8 ijerph-22-01108-t008:** Predictors of supportive supervision satisfaction among the health workforce in 8 African countries.

Health Workforce:		OR (Univariable)	OR (Multivariable)
Country Name	Ethiopia	-	-
	Kenya	5.36 (3.32–8.79, *p* < 0.001)	5.02 (2.88–8.89, *p* < 0.001)
	Malawi	2.00 (1.23–3.25, *p* = 0.005)	1.86 (1.05–3.29, *p* = 0.032)
	Senegal	3.58 (2.32–5.58, *p* < 0.001)	2.78 (1.58–4.95, *p* < 0.001)
	South Sudan	2.82 (1.58–5.10, *p* = 0.001)	1.87 (0.93–3.80, *p* = 0.081)
	Tanzania	5.07 (3.52–7.38, *p* < 0.001)	6.53 (4.19–10.33, *p* < 0.001)
	Uganda	5.49 (3.67–8.31, *p* < 0.001)	3.12 (1.91–5.16, *p* < 0.001)
	Zambia	6.21 (3.57–11.09, *p* < 0.001)	3.03 (1.56–6.03, *p* = 0.001)
Length of Service (Years)	Mean (SD)	1.02 (1.01–1.03, *p* = 0.001)	0.99 (0.97–1.02, *p* = 0.619)
Service Length in Facility (Years)	Mean (SD)	1.04 (1.03–1.06, *p* < 0.001)	1.04 (1.01–1.07, *p* = 0.003)
Respondent Sex	Female	-	-
	Male	0.86 (0.70–1.05, *p* = 0.135)	0.89 (0.70–1.13, *p* = 0.326)
Respondent Age	Mean (SD)	1.02 (1.01–1.03, *p* < 0.001)	1.00 (0.98–1.01, *p* = 0.606)
Cadre	Community Based	-	-
	Facility Based	0.67 (0.55–0.82, *p* < 0.001)	1.08 (0.80–1.46, *p* = 0.601)
Employee–Employer Satisfaction	not satisfactory	-	-
	satisfactory	3.22 (2.59–4.00, *p* < 0.001)	3.10 (2.34–4.11, *p* < 0.001)
Remuneration Satisfaction	not satisfactory	-	-
	satisfactory	1.87 (1.48–2.37, *p* < 0.001)	1.15 (0.83–1.60, *p* = 0.410)
Professional-Development Satisfaction	not satisfactory	-	-
	satisfactory	4.63 (3.72–5.80, *p* < 0.001)	3.55 (2.76–4.59, *p* < 0.001)
Physical-Environment Satisfaction	not satisfactory	-	-
	satisfactory	3.06 (2.39–3.95, *p* < 0.001)	2.12 (1.57–2.86, *p* < 0.001)

OR = Odds Ratio, SD = Standard Deviation.

## Data Availability

Data is available upon reasonable request.

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
