# Peer review of "Predictors of Health-Workforce Job Satisfaction in Primary Care Settings: Insights from a Cross-Sectional Multi-Country Study in Eight African Countries"

_ijerph, 2025, doi:10.3390/ijerph22071108_

Round 1

Reviewer 1 Report

Comments and Suggestions for Authors

This cross-sectional study investigates predictors of job satisfaction among 1,711 health workers in eight sub-Saharan African countries. It assesses satisfaction across five dimensions: employer-employee relationships, remuneration and recognition, professional development, physical work environment, and supportive supervision.

This is an interesting work but there are some issues in the methodology and data validation.

  1. The manuscript does not report any validation of the survey tool. There is no reference to internal consistency checks such as Cronbach’s alpha, nor evidence of prior use or pretesting of the tool.

  1. The sampling methodo lacks details. It appears that health workers were recruited based on their presence at the facility during data collection, but it is not stated whether randomization or systematic sampling was applied. The authors must clarify whether the sample is representative, and whether bias (city or rural locations, weekdays or weekend, day time or nightshift)  may have been introduced.

  1. There is no information on how many participants were approached versus how many completed the survey. Without response rate data, the potential for non-response bias cannot be evaluated. Likewise while the authors report the number of participants from each country and sector, the manuscript does not indicate what percentage they represent of the total health workforce in those areas making it difficult to assess the generalizability of the study.
  2. The Discussion references a few previous studies, but comparisons with existing findings are limited and somewhat superficial. The discussion should more robustly contextualize results in relation to regional and international literature on job satisfaction in health workforces.

Author Response

Point by Point Response

Comments 1: The manuscript does not report any validation of the survey tool. There is no reference to internal consistency checks such as Cronbach’s alpha, nor evidence of prior use or pretesting of the tool

Response 1: Thank you for pointing this out. We agree with this comment. Therefore, we have updated under section 2.3- data collection.

“Tools were collaboratively reviewed by countries and aligned to fit context, questions by each dimension was drafted from existing literature and practice to ensure validity. Main tool in English version was then translated to Kiswahili for Kenya & Tanzania, Arabic for South Sudan, Amharic for Ethiopia, French for Senegal, Chichewa for Zambia and Malawi. Each country conducted standardized training for engaged trained research assistants and embarked on pretest of tools to ensure reliability in data collection, after piloting of tools. Teams converged after pretest exercise to discuss revisions required based on pretest, updates were incorporated to the tools and KoboCollect forms before actual data collection.”

Comments 2: The sampling methods lacks details. It appears that health workers were recruited based on their presence at the facility during data collection, but it is not stated whether randomization or systematic sampling was applied. The authors must clarify whether the sample is representative, and whether bias (city or rural locations, weekdays or weekend, day time or nightshift) may have been introduced.

Response 2: Agreed. We have, accordingly, revise the section to emphasize the “[ Health worker sample was distributed equally across health facilities samples, to get a probable number of persons to interview per facility. Health workers at facility present during day of interviewed were considered for survey where random selection was employed to pick the number required. It should be noted that some smaller facilities may have had fewer staff than the number needed, which meant all are selected and more selected in larger facilities.]”

Comments 3: There is no information on how many participants were approached versus how many completed the survey. Without response rate data, the potential for non-response bias cannot be evaluated. Likewise while the authors report the number of participants from each country and sector, the manuscript does not indicate what percentage they represent of the total health workforce in those areas making it difficult to assess the generalizability of the study.

Response 3: Agree with the comment raised. We have updated table 1 in the manuscript to include the response rate which was 100% in each of the country based on the populations targeted

Comments 4: The Discussion references a few previous studies, but comparisons with existing findings are limited and somewhat superficial. The discussion should more robustly contextualize results in relation to regional and international literature on job satisfaction in health workforces.

Response 4: Agree. The discussion has been updated to be more robust

Reviewer 2 Report

Comments and Suggestions for Authors

Title/Abstract

  • Title is concise, but consider mentioning “A Cross-Sectional Multi-Country Survey” to give immediate clarity on the design.
  • Add a single, clear sentence defining the importance of job satisfaction in sub-Saharan Africa to the background
  • Method: Ensure that the mention of sampling methodology (“cross-sectional study surveyed 1,711 health workers”) is precise. Also, clarify how the five dimensions were measured (Likert scale, scoring).
  • Key Results: You present interesting inter-country comparisons, but it would help to highlight the major factor for which country was significantly distinct (e.g., “Tanzania had the lowest satisfaction with employer, 16%”).
  • Conclusion: The abstract conclusion is good. Potentially, add one practical recommendation or future direction within the abstract’s limit.

Introduction

  •  You provide a clear rationale about the importance of job satisfaction in health workers and situate it well within sub-Saharan Africa. However, references about broader global job satisfaction findings (such as Gallup data) could be tightened. Instead of multiple general references, focus on fewer but specific references that reflect job satisfaction data in healthcare.
  • The introduction does not explicitly highlight the knowledge gap for multi-country data on job satisfaction in Africa (beyond listing each country’s broad challenges). Consider explicitly stating: “Little comparative evidence across sub-Saharan Africa exists that simultaneously examines multiple satisfaction domains, this study addresses that gap.”
  • Study Aim: Stated well. Possibly elaborate on how the findings will be beneficial for “policy relevance.” A short statement about how the eight-country scope was chosen (beyond mentioning Amref presence) would make it more compelling.

Materials and Methods

  • Study Design: The rationale behind combining eight countries requires more justification. Are they intended to represent Eastern, Western, and Southern African contexts equally, or is it purely convenience because of project coverage? Clarify.
  • Sampling and Sample Size: The formula is given in narrative form, but including it as a reference or an appendix might help. Also, the description of the two-stage cluster sampling is minimal; please detail how you selected facilities and then how you selected respondents within them. You mention “(Facility in-charge and 2 staff per facility),” but note that your total number per country sometimes exceeded that straightforward ratio, explain any purposeful oversampling or non-responses.
  • Ethical Considerations: The text does mention IRB approvals for some countries, “exempt” for others. Some clarity is needed about which local boards granted these exemptions and under what conditions. If any countries used only local administrative clearance, clarify.
  • Data Collection Tools: The validity and reliability of the tool are not described. If a pilot test was done, mention how results were used to refine items, especially for cross-cultural adaptation. Also clarify if translations (French, local languages) were used in Senegal or other places.
  • Variables: For each of the five satisfaction dimensions, define the scoring range more explicitly in the methods (e.g., “Scores from 1 to 5 on a Likert scale, aggregated or not?). It is somewhat clarified in the results, but the method should also specify how you converted them to a final “satisfactory vs. not satisfactory.”

Results

  • Demographic Presentation: Authors should provide some clarity on how “Median length of service” was derived is needed (did you show mean in the regression?). Minor consistency check: you mention “Mean (SD) in the predictor tables” but earlier mention “Median (IQR)” in text. Align these approaches.
  • Subgroup Analyses: Good job presenting separate logistic models for the five key satisfaction domains. However, the tables are quite large. It may help to reorganize them for better readability particularly by merging some repeated columns or clarifying each variable’s reference category.
  • Inter-Country Comparisons: The short narratives about high vs. low satisfaction in certain countries are valuable. Some items read repetitively. Suggest linking them with the discussion to avoid duplication.
  • Missing or Inconsistent Indication of p-values: In some results paragraphs, you mention significance (p < 0.001 or p=0.002), whereas in others you just mention “lowest at 16%” without referencing a test or confidence interval. If you tested differences across countries, mention the approach (e.g., a global chi-square or logistic model).
  • Interpretation in Results: The results section at times drifts into interpretative statements (e.g., “... indicating potential employer engagement issues in Tanzania”); consider shifting this interpretive tone to Discussion.

 Discussion

  • Structure: The Discussion is suitably thematically organized following your five domains, but at times merges with the results. A more standard approach is (1) Major findings, (2) Comparison with existing studies, (3) Strengths/limitations, (4) Practical implications, (5) Conclusion.
  • Comparison to Prior Literature: Good references to a few studies in Ethiopia, Malawi, or other. Yet a more explicit alignment with recognized frameworks (e.g., Herzberg’s) would make the discussion stronger. For instance, motivational factors (recognition, professional development) vs. hygiene factors (salary, environment, supervision).
  • Granularity: The discussion occasionally lumps all LMIC contexts together. Where possible, highlight any unique aspects discovered, e.g., “Zambia’s higher satisfaction might be due to X.” But ensure it is tied to references or known policies in that context.
  • Limitations: You mention standard constraints cross-sectional design, potential non-response. Expand on potential reporting biases. The large multi-country scope might also hamper direct local contextual nuance. Also, highlight that the proportion of staff from each cadre might differ across countries did that confound the results?

Conclusions

  • Conciseness: The conclusion is concise. Possibly underscore one or two key, domain-specific interventions (like “improving salary structures or investing in robust supportive supervision”).
  • Forward-looking: Add a final line on next steps: e.g., how policy could integrate these findings or any potential for repeated surveys to track improvements over time.

References

  • In general, references are relevant, but some references for global contexts (like Gallup’s global stats) could be condensed or replaced with more Africa-specific workforce satisfaction data if available. Also ensure you consistently follow the journal’s format (e.g., consistent bracket numbering or numbering style).
Comments on the Quality of English Language

The manuscript’s English is generally acceptable. Some minor grammatical inconsistencies or repetitive phrases appear (e.g., “eight African countries” repeated). A careful copyedit would help.

Author Response

Response to Reviewer 2 comments

Comments 1: Title is concise, but consider mentioning “A Cross-Sectional Multi-Country Survey” to give immediate clarity on the design

Response 1: The title has been revised to include the study design

Comments 2: Add a single, clear sentence defining the importance of job satisfaction in sub-Saharan Africa to the background

Response 2: Agreed. We have included this sentence in the background of the abstract  highlighting importance of Job satisfaction. Job satisfaction in sub-Saharan Africa is crucial as it directly impacts employee productivity, retention, and overall economic growth, fostering a motivated workforce that drives development in the region

Comments 3: Method: Ensure that the mention of sampling methodology (“cross-sectional study surveyed 1,711 health workers”) is precise. Also, clarify how the five dimensions were measured (Likert scale, scoring).

Response 3: Agreed. A sentence on how the 5 dimensions has been included in the abstract

Comments 4: Key Results: You present interesting inter-country comparisons, but it would help to highlight the major factor for which country was significantly distinct (e.g., “Tanzania had the lowest satisfaction with employer, 16%”).

Response 4: Agreed. The results have been revised to reflect the major factors by Countries.

Comments 4: Conclusion: The abstract conclusion is good. Potentially, add one practical recommendation or future direction within the abstract’s limit.

Response 4: Agreed. We have, accordingly, modified the conclusion in the abstract to bring out future implications of the findings

Introduction

Comments 5: You provide a clear rationale about the importance of job satisfaction in health workers and situate it well within sub-Saharan Africa. However, references about broader global job satisfaction findings (such as Gallup data) could be tightened. Instead of multiple general references, focus on fewer but specific references that reflect job satisfaction data in healthcare

Response 5: Agreed. We have, accordingly, revised the introduction to include more references that reflect job satisfaction data in healthcare . New material has been introduced on pages 2-3, lines 64 – 75.

A study by Muthuri and colleagues done in the East Africa Community (EAC) indicates that there are individual, organizational/structural and societal determinants of healthcare workers’ motivation reported by the health workforce [(Muthuri et al., 2020)]. Muthuri and colleagues mention that barriers to motivation in their systematic review included lack of or inadequate monetary support, favouritism, critical ism, critical shortage of skilled healthcare professionals leading to heavy workload, and unrealistic expectations from management and government (Muthuri et al., 2020).

Blaauw, et. al. (2013) in their study reports that 82.3% of respondents in Tanzania were satisfied with their jobs, compared to 71.0% in Malawi, and 52.1% in South Africa. In all three countries, health workers were most satisfied with their job variety and the opportunity to fully utilise their abilities.

“[updated text in the manuscript if necessary]”

Comments 6: The introduction does not explicitly highlight the knowledge gap for multi-country data on job satisfaction in Africa (beyond listing each country’s broad challenges). Consider explicitly stating: “Little comparative evidence across sub-Saharan Africa exists that simultaneously examines multiple satisfaction domains, this study addresses that gap.”

Response 6: Agree. We have modified section 1, Introduction line 77-79..to emphasize this point.

Comments 7: Study Aim: Stated well. Possibly elaborate on how the findings will be beneficial for “policy relevance.” A short statement about how the eight-country scope was chosen (beyond mentioning Amref presence) would make it more compelling.

Response 7: Agreed. In response, we have included a paragraph emphasizing the policy relevance of the study and the necessity for country-specific, tailored interventions. These interventions will guide Amref's implementation across multiple countries, ensuring that each nation's unique challenges are addressed effectively and that the interventions are aligned with the specific needs of their health workforce.

Materials and Methods

Comments 8: Study Design: The rationale behind combining eight countries requires more justification. Are they intended to represent Eastern, Western, and Southern African contexts equally, or is it purely convenience because of project coverage? Clarify.

Response 8: Agree. We have, accordingly, included a rationale for selecting the 8 countries in addition to Amref presence with Health workforce interventions

Comments 9: Sampling and Sample Size: The formula is given in narrative form, but including it as a reference or an appendix might help. Also, the description of the two-stage cluster sampling is minimal; please detail how you selected facilities and then how you selected respondents within them. You mention “(Facility in-charge and 2 staff per facility),” but note that your total number per country sometimes exceeded that straightforward ratio, explain any purposeful oversampling or non-responses.

Response 9: Agreed. We have, accordingly, revised the sampling strategy for clarity on the two stage sampling and more details on sampling procedures for healthcare workers in the facilities. Details are included in section 2.4

Comments 10: Ethical Considerations: The text does mention IRB approvals for some countries, “exempt” for others. Some clarity is needed about which local boards granted these exemptions and under what conditions. If any countries used only local administrative clearance, clarify.

Response 10: Agreed. Waiver letters were granted by Ethics Boards in Malawi and Senegal mainly because they did not meet a threshold for committees’ full review being an evaluation. The details are included in the IRB statement section and a copy of waiver letter attached

Comments 11: Data Collection Tools: The validity and reliability of the tool are not described. If a pilot test was done, mention how results were used to refine items, especially for cross-cultural adaptation. Also clarify if translations (French, local languages) were used in Senegal or other places.

Response 11: Agreed. We have, accordingly, revised section 2.3 , data collection, lines 197 to 205..to emphasize this point. “[ Tools were collaboratively reviewed by countries and aligned to fit context, questions by each dimension was drafted from existing literature and practice to ensure validity. Main tool in English version was then translated to Kiswahili for Kenya & Tanzania, Arabic for South Sudan, Amharic for Ethiopia, French for Senegal, Chichewa for Zambia and Malawi. Each country conducted standardized trainings for engaged trained research assistants and embarked on pretest of tools to ensure reliability in data collection, after piloting of tools. Teams converged after pretest exercise to discuss revisions required based on pretest, updates were incorporated to the tools including and KoboCollect forms before actual data collection.

Comments 12: Variables: For each of the five satisfaction dimensions, define the scoring range more explicitly in the methods (e.g., “Scores from 1 to 5 on a Likert scale, aggregated or not?). It is somewhat clarified in the results, but the method should also specify how you converted them to a final “satisfactory vs. not satisfactory.”

Response 12: Agreed. Very dissatisfied =1, disatisfied =2, neural =3, satisfied=4, very satisfied=5. This was collapsed into two categories; very dissatisfied and dissatisfied were combined into 1 category called ‘not satisfactory’. Satisfied and very satisfied were collapsed as ‘satisfactory’. Poor and neutral were excluded from the analysis hence the dichotomy of satisfactory and not-satisfactory

Results

Comments 13: Demographic Presentation: Authors should provide some clarity on how “Median length of service” was derived is needed (did you show mean in the regression?). Minor consistency check: you mention “Mean (SD) in the predictor tables” but earlier mention “Median (IQR)” in text. Align these approaches.

Response 13: Agreed. We clarify that while medians and IQRs are better for summarizing skewed data in descriptive tables, modelling requires the raw (often mean-based) scale to accurately estimate and interpret effects. Odds Ratios (ORs) for continuous variables reflect the change in the odds of the outcome per unit increase in the predictor. This change is based on the actual scale of the variable, not its median value. Therefore, using medians in this context wouldn't be appropriate because:

  1. ORs are calculated per unit change, not per change in median—so the median isn’t relevant to the interpretation of the regression coefficient.
  2. Logistic regression estimates a linear relationship between the predictor and the log-odds of the outcome. This estimation is based on the actual distribution of values, typically around the mean, not the median.

Comments 14: Subgroup Analyses: Good job presenting separate logistic models for the five key satisfaction domains. However, the tables are quite large. It may help to reorganize them for better readability particularly by merging some repeated columns or clarifying each variable’s reference category

Response 14: Disagreed. All columns for each of the dimensions are relevant in understanding the modeling

Comments 15: Inter-Country Comparisons: The short narratives about high vs. low satisfaction in certain countries are valuable. Some items read repetitively. Suggest linking them with the discussion to avoid duplication

Response 15: Disagree. Intercountry descriptions are necessary. Areas with duplications have been revised

Comments 16: Missing or Inconsistent Indication of p-values: In some results paragraphs, you mention significance (p < 0.001 or p=0.002), whereas in others you just mention “lowest at 16%” without referencing a test or confidence interval. If you tested differences across countries, mention the approach (e.g., a global chi-square or logistic model).

Response 16: Agreed. We have, included pvalues in sentences that were missing and these are highlighted

Comments 17: Interpretation in Results: The results section at times drifts into interpretative statements (e.g., “... indicating potential employer engagement issues in Tanzania”); consider shifting this interpretive tone to Discussion

Response 17: Agree. Misleading statements have been revised to include true picture of the results

Discussion

Comments 18: Structure: The Discussion is suitably thematically organized following your five domains, but at times merges with the results. A more standard approach is (1) Major findings, (2) Comparison with existing studies, (3) Strengths/limitations, (4) Practical implications, (5) Conclusion.

Response 18: Agreed.  The entire discussion session has been revised to follow the standard approach

Comments 19: Comparison to Prior Literature: Good references to a few studies in Ethiopia, Malawi, or other. Yet a more explicit alignment with recognized frameworks (e.g., Herzberg’s) would make the discussion stronger. For instance, motivational factors (recognition, professional development) vs. hygiene factors (salary, environment, supervision)

Response 19: Agreed, the discussion section has been revised to include alignment to the Herzbergs framework

Comments 20: Granularity: The discussion occasionally lumps all LMIC contexts together. Where possible, highlight any unique aspects discovered, e.g., “Zambia’s higher satisfaction might be due to X.” But ensure it is tied to references or known policies in that context.

Response 20: Agreed. The discussion section has been revised to bring out and discuss major findings that bring out unique aspects

Comments 21: Limitations: You mention standard constraints cross-sectional design, potential non-response. Expand on potential reporting biases. The large multi-country scope might also hamper direct local contextual nuance. Also, highlight that the proportion of staff from each cadre might differ across countries did that confound the results?

Response 21: Agreed. Study limitations have been included as guided

Conclusions

Comments 22: Conciseness: The conclusion is concise. Possibly underscore one or two key, domain-specific interventions (like “improving salary structures or investing in robust supportive supervision”).

Response 22: Agreed and conclusions have been revised to reflect conciseness

Comments 23: Forward-looking: Add a final line on next steps: e.g., how policy could integrate these findings or any potential for repeated surveys to track improvements over time.

Response 23: Agreed. Policy implications have been included

References

Comments 24: In general, references are relevant, but some references for global contexts (like Gallup’s global stats) could be condensed or replaced with more Africa-specific workforce satisfaction data if available. Also ensure you consistently follow the journal’s format (e.g., consistent bracket numbering or numbering style).

Response 24: Agreed. Correct referencing has been incorporated based on the journal guidelines

Round 2

Reviewer 1 Report

Comments and Suggestions for Authors

The authors have revised and improved the manuscript based on my comments and suggestions, particularly in relation to the sampling methods, response rate, and tool validation.

However, there are two points that are partially addressed:

  1. Validation of the survey tool, where the authors should explain if the tool was validated and mention of statistical indicators tests like Cronbach's alpha.

  1. Discussion of existing literature, where the authors should provide concrete examples of how they have expanded the discussion and compared their findings to regional and international studies.

Author Response

The authors have revised and improved the manuscript based on my comments and suggestions, particularly in relation to the sampling methods, response rate, and tool validation. However, there are two points that are partially addressed: 

Comment 1: Validation of the survey tool, where the authors should explain if the tool was validated and mention statistical indicators tests like Cronbach's alpha. 

Response:  The following activities were undertaken to validate the tool however, statistical indicators tests like Cronbach’s alpha was not done

  • Co creation of the tools with all country representations (Amref team) who bring experience and understanding of governments/ Ministries of health, and alignment to similar work done in the past
  • Tools were reviewed and contextualized at country level
  • Tools were pretested at country level and feedback incorporated into the tools before the survey

Comment 2: Discussion of existing literature, where the authors should provide concrete examples of how they have expanded the discussion and compared their findings to regional and international studies.

Response: We appreciate the reviewer’s feedback and have expanded our discussion to more clearly demonstrate how our findings contribute to existing literature. Specifically:

Regional Comparisons: Our study utilizes standardized tools to compare HRH characteristics across eight sub-Saharan African countries. For instance, we contextualize workforce stability (e.g., longer median tenure in Zambia vs. shorter in Ethiopia) and gender composition (e.g., Zambia's 72% female workforce vs. Malawi's 43%) within regional dynamics and global trends.

Integration of Global Literature: We draw on WHO data and international studies (e.g., Kok et al., 2020; WHO 2022) to frame workforce attrition and migration challenges, demonstrating how our findings align with global evidence on health worker turnover and retention.
Theoretical Framing: We applied Herzberg’s Two-Factor Theory of Motivation to interpret job satisfaction dimensions—linking extrinsic and intrinsic factors to actionable policy recommendations.

Comparative Analysis: Through multivariable regression, we highlighted significant variations in job satisfaction across countries (e.g., Zambia vs. Tanzania), reinforcing the need for tailored, context-specific HRH strategies.

Reviewer 2 Report

Comments and Suggestions for Authors

Brief Comments for the Authors

  • Your point-by-point responses address all methodological and presentation concerns; thank you for the careful revisions.
  • To enhance readability, compress large results tables or relocate them to supplementary files; in-text refer only to key ORs.
  • Verify that all references follow IJERPH style (journal full names, DOI on same line, punctuation).
  • Add software credit to Figure 1 caption if applicable.

Remaining minor issues are editorial rather than substantive:

  1. Language/formatting – a final copy-edit is still needed for scattered grammatical slips (e.g., “critical ism” → “criticism”; inconsistent spacing before %).
  2. Large tables – consider moving the full multivariable models to Online Supplementary Material and keeping a trimmed version (variables with p < 0.10) in the main text.
  3. Reference style – a few DOIs and journal titles need to be conformed to IJERPH style; ensure sequential numbering after edits.
  4. Figure 1 credit – add “Created with QGIS 3.34” (or similar) if produced with GIS software.
Comments on the Quality of English Language
  • Please run the manuscript through a professional copy-edit (or a final grammar-check tool) to remove small typographical errors and harmonise percentage/decimal formatting.

Author Response

  • Your point-by-point responses address all methodological and presentation concerns; thank you for the careful revisions.
  • To enhance readability, compress large results tables or relocate them to supplementary files; in-text refer only to key ORs.
  • Verify that all references follow IJERPH style (journal full names, DOI on same line, punctuation).
  • Add software credit to Figure 1 caption if applicable.

Remaining minor issues are editorial rather than substantive:

  1. Language/formatting – a final copy-edit is still needed for scattered grammatical slips (e.g., “critical ism” → “criticism”; inconsistent spacing before %). - reviewed appropriately
  2. Large tables – consider moving the full multivariable models to Online Supplementary Material and keeping a trimmed version (variables with p < 0.10) in the main text. - trimmed the tabled appropriately
  3. Reference style – a few DOIs and journal titles need to be conformed to IJERPH style; ensure sequential numbering after edits. - updated references
  4. Figure 1 credit – add “Created with QGIS 3.34” (or similar) if produced with GIS software. - updated on figure 1